# Optical Flow-Aware-Based Multi-Modal Fusion Network for Violence Detection

**DOI:** 10.3390/e24070939

**Published:** 2022-07-06

**Authors:** Yang Xiao, Guxue Gao, Liejun Wang, Huicheng Lai

**Affiliations:** Xinjiang Key Laboratory of Signal Detection and Processing, College of Information Science and Engineering, Xinjiang University, Urumqi 830046, China; xiaoyang@stu.xju.edu.cn (Y.X.); gaoyangshuang123@stu.xju.edu.cn (G.G.); wljxju@xju.edu.cn (L.W.)

**Keywords:** violence detection, multi-modal fusion, adaptive fusion, optical flow-aware

## Abstract

Violence detection aims to locate violent content in video frames. Improving the accuracy of violence detection is of great importance for security. However, the current methods do not make full use of the multi-modal vision and audio information, which affects the accuracy of violence detection. We found that the violence detection accuracy of different kinds of videos is related to the change of optical flow. With this in mind, we propose an optical flow-aware-based multi-modal fusion network (OAMFN) for violence detection. Specifically, we use three different fusion strategies to fully integrate multi-modal features. First, the main branch concatenates RGB features and audio features and the optical flow branch concatenates optical flow features with RGB features and audio features, respectively. Then, the cross-modal information fusion module integrates the features of different combinations and applies weights to them to capture cross-modal information in audio and video. After that, the channel attention module extracts valuable information by weighting the integration features. Furthermore, an optical flow-aware-based score fusion strategy is introduced to fuse features of different modalities from two branches. Compared with methods on the XD-Violence dataset, our multi-modal fusion network yields APs that are 83.09% and 1.4% higher than those of the state-of-the-art methods in offline detection, and 78.09% and 4.42% higher than those of the state-of-the-art methods in online detection.

## 1. Introduction

Video surveillance systems are being installed all over the world. However, manual detection has many problems, such as high costs and a low efficiency, which cannot meet the needs of the public. With the development of computer vision, more and more researchers have begun to pay attention to video surveillance systems. For example, traffic videos are obtained by monitoring cameras at various locations on the road, and the current monitoring of the videos is carried out to complete the tracking tasks of accident determination and specific behaviors [1]. Moreover, with the economy growing, the frequent occurrence of public safety accidents has become increasingly serious. Violent events, such as fights, robberies, and other unusual incidents occur frequently. As a result, violence detection technology has been developed. Violence detection technology can be applied to a variety of scenarios. For example, on the vehicle internet, through the detection of traffic accidents, real-time alarms can be given to reduce damage.

Indeed, most of the previous violence detection networks were based on visual information [2,3,4,5,6,7,8,9,10,11,12,13,14]. Hu et al. [10] used graph neural networks (GCNs) to correct noise labels through feature similarity and time continuity. The corrected labels were used to train an action classifier based on supervised learning to improve the accuracy of abnormal event locations in the time dimension. In addition, the attention mechanism was also used in previous violence detection algorithms. Zhu et al. [9] proposed the use of an attention module to strengthen the network learning of motion features. These networks neglected the important information contained in the audio, which made the accuracy of the anomaly detection networks in certain situations (such as a quarrel between two people) not ideal. In addition, due to the diversity and the complexity of the violence, the current violence detection algorithm has high accuracy in simple scenes, but the accuracy is not ideal in scenes with dense objects, complex backgrounds, and mutual occlusion.

To address the above problems, some researchers have attempted to utilize multi-modal information [15,16,17,18,19,20,21,22], such as audio and text information, to improve the performance of violence detection.

Nam et al. [15] proposed, for the first time, that the algorithm of violent behavior detection was based on the joint training of audio, color, and motion characteristics to obtain a classifier, which was used to grade and classify the violence of movies in 1998. Theodoros et al. [16] defined the classification of audio and visual features, used a Bayesian network to classify the features, and then sent the audio and visual features to a KNN classifier for classification. Lin et al. [17] combined audio and visual classifiers in collaborative training to detect video anomalies from the perspectives of video and audio. Giannakopoulos et al. [18] fused audio, image and text modals and classified them on a KNN classifier to form a two-dimensional classification problem in a nine-dimensional feature space. Zou et al. [19] adopted a multistep method on the basis of predecessors. First, text information was used to build a preclassifier to select potentially exceptional fragments. Second, the SVM classifier combined visual and audio information to divide the potential abnormal segments into “abnormal” or “non-abnormal”. Cristani et al. [20] extracted the features of the two signals and used the two features to train the KNN classifier for classification and recognition. Zajde et al. [21] proposed a dynamic Bayesian network (DBN) to extract and integrate the underlying audio and visual features, and applied it to the intelligent video surveillance system, CASSANDRA. Gong et al. [23] designed a semisupervised cross feature learning algorithm to extract the underlying audio and visual information.

The above methods are based on handcrafted features, but complex behaviors cannot be completely expressed by relatively single handcrafted features. Therefore, Wu et al. [22] built a large-scale dataset named XD-Violence and used a neural network to extract depth features to fill the gap in the violence detection dataset. Additionally, based on Hu et al. [10], they improved the algorithm and proposed a neural network with three parallel branches to capture the different relationships between video clips. They added audio information as inputs to further improve the detection accuracy. Wu et al. [24] proposed a method composed of causal temporal relation (CTR), classifier (CL), compactness (CP), and dispersion (DP) modules to explore causal temporal relations and feature discrimination ability in a local scope, so as to solve the problems caused by the lack of temporal relation modeling and feature discrimination in previous methods. Li et al. [25] proposed a multi sequence learning (MSL) method and a hinge-based MSL ranking loss method. By using a sequence composed of multiple segments as an optimization unit, they reduced the probability of selection errors during training. Pang et al. [26] further improved the algorithm on the basis of Wu et al. [22], who focused on fusing audio and visual information. First, weighted features are used to generate effective features under the guidance of audio and visual information. Second, a fusion module is added. Visual and audio information are fused into features based on a bilinear pool mechanism. Finally, a mutual learning module is added to make the model learn visual information from another neural network with a different structure.

Unlike Pang’s method [26], we found that the violence detection accuracy of different kinds of videos is related to the change of optical flow. Therefore, our network added the optical flow feature to extract the motion features of objects and effectively solve the problems of short duration and weak action for the task of violence detection. In addition, we designed the optical flow-aware-based score fusion strategy to fuse the branch with optical flow features and the branch without optical flow features, which can control the influence of optical flow features on violence detection. First, we concatenate different kinds of modalities, such as RGB–optical flow, optical flow–audio, and audio–RGB, to form three branches. After weighting each branch to capture the cross-modal information from each modality, we concatenate the RGB–optical flow branch and the optical flow–audio branch to form the optical flow branch. At the same time, the audio–RGB branch is named the main branch. Second, the channel attention module (CAM) captures the features in two branches that are more helpful for classification. Then, the feature map is down-sampled by the transition layer to reduce the complexity of the module. Finally, the prediction module captures the distance relationship between two positions and predicts the scores of online detection and offline detection in two branches. The optical flow-aware network then weights the scores of the main branch and the optical flow branch via a gate function defined over the optical flow value.

In summary, this work has the following four main contributions:We propose a novel two-branch optical flow-aware-based multi-modal fusion network for violence detection, which integrates audio features, the optical flow features, and the RGB features into a unified framework;We introduce three different fusion strategies for extracting important information and suppressing unnecessary audio and visual information, which includes an input fusion strategy, attention-based halfway fusion strategy, and optical flow-aware-based score fusion strategy;We propose an optical flow-aware score weighting mechanism to control the contributions of the main branch and the optical flow branch under different optical flow conditions and to boost the AP performance of violence detection;A novel cross-modal information fusion module (CIFM) and novel channel attention module are proposed to weight the combined feature, which can extract useful information from features while eliminating useless information, such as redundancies and noise.

## 2. Related Work

### 2.1. Multi-Modal Fusion Strategies

Li et al. [27] studied six different multi-modal fusion networks. The six fusion strategies are input fusion, early fusion, halfway fusion, late fusion, and two kinds of score fusion strategies. In the early fusion network, they concatenated color and thermal modalities directly. In the early fusion network, they integrated color and thermal modalities after the first convolution block. In the halfway fusion network, they connected the features of the two modals through the feature map, and then used NiN to reduce the dimension, then connecting the color and thermal modalities. In the late fusion network, they connected the feature maps of the two modal sub-networks after the last convolution block. The score fusion network can be seen as a cascade design of two sub-networks; the detection results are obtained by combining the two-stage detection confidence scores with the same weight of 0.5.

### 2.2. Multi-Modal Fusion Methods

Most of the existing fusion strategies (direct fusion, bilinear pooling-based fusion [28]) either cannot make full use of cross-modal information or the amount of calculation is too large. An attention-based fusion strategy can effectively avoid the above problems. The attention mechanism is a technology widely used in multi-modal fusion; in particular, the attention mechanism is often used for mapping between modals.

#### 2.2.1. Single Modal Attention

A number of studies [29,30,31] have shown that embedding an attention module into image classification, object detection, and other tasks can result in a substantial performance improvement. Li et al. [32] designed the fusion net to select the k most-representative feature maps to realize the adaptive fusion of different modals, while avoiding redundant noise. Gao et al. [33] weighted the modals to make the network focus on more favorable fields to effectively integrate different modals. Zhang et al. [34] took the feature maps from two-stream Siamese networks as inputs and weighted the features through the weight generation sub-network to obtain the additional information between modals. Then, the enhanced features were obtained by using cross-modal residual connections, and finally, these features were concatenated. Lu et al. [35] designed an instance adapter to use two fully connected layers for each modal, and then predicted the modal weight to realize the quality-aware fusion of different modals.

#### 2.2.2. Cross-Modal Attention

Cross-modal mechanisms have increasingly become the focus of multi-modal fusion. Dou et al. [36] used a cross-attention mechanism in their co-attention module to realize cross-modal interaction. Liu et al. [37] compressed the features of the two modals through a cross-modal encoder, and then used multi-head attention to transfer the expanded features back to each modal. From the realization of modal interaction, the two-stage cross-modal feature propagation can enhance the audio and visual features and eliminate the noise information. Badamdorj et al. [38] fused the bimodal-attention module to extract the interaction between the audio and visual features, so as to improve the accuracy of highlight detection. Jiang et al. [39] designed a cross-modal fusion module, that is, a multi-level cross-spatial attention module. Firstly, the features of each encoder are transmitted to the cross-modal fusion module to calculate the cross-modal attention, and then these weighted features are connected, respectively, spliced, and mapped back to the original dimension. Hendricks et al. [40] proposed two kinds of fusion attention, namely, merged attention and co-attention. Dou et al. [36] studied these two kinds of attention. In the merged attention module, they concatenated text features and visual features, and then fed them into the transformer block. In the co-attention module, text features and visual features were fed into different transformer blocks separately; then, they used the cross-attention module to conduct “cross talk”.

## 3. Proposed Work

In this section, we describe our proposed violence detection network based on an optical flow-aware weighting mechanism and multi-modal fusion (OAMFN) in detail. The overall framework is shown in Figure 1, which consists of three parts: the cross-modal information fusion module, the channel attention module, and the prediction module. Section 3.1 describes the details of the cross-modal information fusion module used to capture the cross-modal information and fuse the multi-modal features. Section 3.2 introduces the details of the channel attention module and Section 3.3 describes the composition and function of the prediction module.

### 3.1. The Cross-Modal Information Fusion Module

Both visual and audio features have noisy information and spatial temporal redundancy [36], which interferes with violence detection. However, cross-modal feature propagation can enhance audio and visual features and suppress noise information.

The cross-modal information fusion module (CIFM) integrates the features of different combinations and weights them to capture the cross-modal information in audio and video. As shown in Figure 1, for the inputs of OAMFN, we denote $x_{rgb}\in R^{T\times D}$ as RGB features, $x_{flow}\in R^{T\times D}$ as optical flow features, and $x_{audio}\in R^{T\times D_{1}}$ as audio features, where *T* is the length of the feature matrix, *D* is the dimensions of the RGB features and the optical flow features, and $D_{1}$ is the dimensions of the audio features.

#### 3.1.1. Modal Combination

At this stage, the CIFM integrates information from three modals. To reduce the amount of calculation at this stage, we adopt the concatenate operation to fuse two kinds of features from three modals as follows:(1)$$x_{rf}=cat\left(x_{rgb},x_{flow}\right)$$
(2)$$x_{ra}=cat\left(x_{rgb},x_{audio}\right)$$
(3)$$x_{fa}=cat\left(x_{flow},x_{audio}\right)$$
where cat is the concatenation operation.

#### 3.1.2. Adaptive Fusion

To capture the cross-modal information from each kind of multi-modal feature, we adopt adaptive weights for these branches as follows:(4)$$w_{rf}=\sigma \left(smu\left(bn\left(avg\left(conv\left(cat\left(x_{rgb},x_{flow}\right)\right)\right)\right)\right)\right)$$
(5)$$w_{ra}=\sigma \left(smu\left(bn\left(avg\left(conv\left(cat\left(x_{rgb},x_{audio}\right)\right)\right)\right)\right)\right)$$
(6)$$w_{fa}=\sigma \left(smu\left(bn\left(avg\left(conv\left(cat\left(x_{flow},x_{audio}\right)\right)\right)\right)\right)\right)$$
where σ is the sigmoid layer used to compute the final weights of different branches, conv denotes a convolutional layer with a kernel size of one, and avg and bn epresent the average pooling and the batch normalization, respectively. Additionally, smu is the smooth maximum unit [41], which is defined as follows:(7)$$\frac{d}{dx}erf\left(x\right)=\frac{2}{\sqrt{\pi }}e^{-x^{2}}$$
where *x* is the input variable.

In this paper, we take the smooth maximum unit (SMU) [41] as the activation function of the network. The function $\left|x\right|$ is nondifferentiable at the origin. Therefore, Biswas et al. [41] used the smooth function to approximate the $\left|x\right|$ function. They found a general approximation formula of the maximum function from the smooth approximation of the $\left|x\right|$ function, which can smoothly approximate the general maxout [42] family, ReLU, leaky ReLU, or its variants, such as Swish, etc. In addition, the author also proves that the GELU function is a special case of the SMU. Experiments show that the SMU is effective in the fields of image classification, object detection, and semantic segmentation.

With the weights $w_{rf}$, $w_{ra}$ and $w_{fa}$, the enhanced multi-modal features are as follows:(8)$$F_{rf}=x_{rf}*w_{rf}$$
(9)$$F_{ra}=x_{ra}*w_{ra}$$
(10)$$F_{fa}=x_{fa}*w_{fa}$$

Finally, we divide the output of the CIFM into two branches, the main branch and the optical flow branch. The output of the optical flow branch is defined as follows:(11)$$F_{opt}=cat\left(F_{rf},F_{fa}\right)$$

The output of the main branch is defined as follows:(12)$$F_{main}=F_{ra}$$

### 3.2. Channel Attention Module

In this section, we describe the details of the channel attention module (CAM). Note that the cross-modal information fusion module has extracted the cross-modal audio and visual information. Generally, channel attention focuses on what is meaningful in input features. As shown in Figure 2, to retain valuable information in $F_{opt}$ and $F_{main}$, we adopt a weighting operation in CAM to make the network pay attention to information which is more useful to improve the accuracy of the prediction.

To realize the mapping of the feature map from “small resolution” to “large resolution”, we take the up-sampling operation. There are three common methods of up-sampling: bilinear interpolation, transposed convolution, and unpooling. In this paper, we adopt the transposed convolution layer. Here, we use $F_{fuse}$ to represent $F_{opt}$ and $F_{main}$. First, the size of the input feature $F_{fuse}$ is expanded by adding zero according to a certain proportion. Then, the convolution kernel is rotated, and the forward convolution is carried out as follows:(13)$$F_{fuse}^{\prime }=smu\left(deconv\left(F_{fuse}\right)\right)$$
where deconv denotes the transposed convolution, which can enrich the information of the feature map. In the transposed convolution layer, the kernel size is nine, the stride is three and the padding is four. To soften the input feature $F_{fuse}$, we add an SMU layer after the transposed convolution layer.

To effectively calculate the channel attention, it is necessary to compress the spatial dimension of the input feature map. For the aggregation of spatial information, the common method is global average pooling, as follows:(14)$$F_{g}=gap\left(F_{fuse}^{\prime }\right)$$
where gap denotes the global average pooling layer, which can regularize the structure of the whole network to prevent overfitting.

To convert features to a new feature space, we adopt conv4 and conv5 with a kernel size of one after the global average pooling layer. Additionally, we add a batch normalization layer to normalize the network output, which can make the gradient larger to avoid the problem of gradient disappearance, as follows:(15)$$F_{b}=smu\left(bn\left(smu\left(conv5\left(smu\left(conv4F_{g}\right)\right)\right)\right)\right)$$
where conv5 and conv4 denote a filter with a kernel size of one.

Finally, the sigmoid function is used to generate weights as follows:(16)$$w_{c}=\sigma \left(F_{b}\right)$$

In general, the outputs of the CAM are defined as follows:(17)$$F_{c}=F_{fuse}*w_{c}$$
where ∗ denotes the element-wise multiplication, and the outputs of the CAM are defined as $F_{c-opt}$ and $F_{c-main}$.

### 3.3. Prediction Module

As shown in Figure 1, the proposed prediction module consists of a main branch and an optical flow branch; each branch contains a backbone and two prediction branches, namely, an offline prediction branch and an online prediction branch. The backbone is mainly used to smooth the weighted features, and it uses the dropout mechanism to enhance the generalizability of the network. In a violence detection system, online detection and offline detection both play an important role. Offline systems are mostly used for monitoring videos and videotapes. Online detection is mostly used to detect online video violence in real time. We discuss each part in detail.

#### 3.3.1. Transition Layer

To soften the fused features and introduce a pooling operation to change the size of the feature map, we propose a transition layer, including a convolution layer and an average pooling layer. The transition layer used in this paper is expressed mathematically as follows:(18)$$F_{t-main}=smu\left(bn\left(avg\left(conv\left(F_{c-main}\right)\right)\right)\right)$$
(19)$$F_{t-opt}=smu\left(bn\left(avg\left(conv\left(F_{c-opt}\right)\right)\right)\right)$$
where conv denotes a convolutional layer with a kernel size of one, avg and bn represent the average pooling and the batch normalization, respectively, and $F_{t-main}$ and $F_{t-opt}$ represent the features in the main branch and the optical flow branch, respectively.

#### 3.3.2. Backbone

To further smooth the weighted features, we add two transition layers to smooth them. In addition, to improve the generalization of the model, we add double dropout layers as follows:(20)$$F_{b}^{\prime }=Drop\left(T\left(F_{c}\right)\right)$$
(21)$$T\left(x\right)=smu\left(bn\left(avg\left(conv\left(x\right)\right)\right)\right)$$
where Drop denotes the dropout layer, and the dropout rate is 0.3. $T\left(x\right)$ is the expression of the transition layer. In general, the outputs of the backbone are defined as follows:(22)$$F_{main}=Drop\left(T\left(F_{main}^{\prime }\right)\right)$$
(23)$$F_{b-opt}=Drop\left(T\left(F_{b-opt}^{\prime }\right)\right)$$

#### 3.3.3. Optical Flow-Aware Weighting

To control the influence of optical flow features under different optical flow conditions, an optical flow-aware-based score fusion strategy is introduced to fuse features of different modalities from two branches. When the optical flow changes greatly, the weight of the optical flow branch should be relatively high, while the weight of the main branch should be relatively low, so as to improve the detection accuracy of the corresponding violence classes.

Hence, we present an optical flow gate which limits the weight of optical flow branches as follows:(24)$$w=\frac{x_{flow}}{1+\mu \exp\left(-\frac{x_{flow}-0.5}{\theta }\right)}$$
where μ and θ denote hyperparameters that can control the influence of the optical flow on the weight.

Finally, we term $w_{opt}=w$ and $w_{main}=1-w$ as the weights for the optical flow-aware-based score fusion, where $w_{main}$ and $w_{opt}$ can control the contributions of the main branch and the optical flow branch under different optical flow conditions.

#### 3.3.4. Offline Detection

However, there is still a challenging problem to be solved. The CAM can only effectively capture local information, but it cannot establish long-term dependence between two positions. Therefore, we take the localized branch of the HL-Net [22] to capture the distance relationship between two positions as follows:(25)$$F_{off}\left(i,j\right)=\exp\left(\frac{-{\left|i-j\right|}^{\beta }}{\alpha }\right)$$
where *i*, *j* denotes the *i*th and *j*th features, and *α* and *β* denote hyperparameters that can control the influence of the distance relationship between two positions.

#### 3.3.5. Online Detection

In addition to offline detection, online detection is also important. It can detect the video in the network and detect the surveillance video in real time, which has great application value.

First, we use a transition layer to smooth the features as follows:(26)$$T\left(x\right)=smu\left(bn\left(avg\left(conv\left(x\right)\right)\right)\right)$$

After the transition layer, there is a 1D convolution layer with the SMU for activation as follows:(27)$$F_{on}=smu\left(conv\left(T\left(x\right)\right)\right)$$
where conv is the 1D convolution layer with a kernel size of one.

Finally, we use a 1D causal revolution layer with a kernel size of five as the classifier for online detection.

#### 3.3.6. Optical Flow-Aware Score Fusion

Each branch generates two outputs: an offline detection score $S_{off}$ and an online detection score $S_{on}$. Then, given $S_{off-main}$ and $S_{on-main}$ from the main branch and $S_{off-opt}$ and $S_{on-opt}$, we obtain the final violence detection score:(28)$$S_{off-final}=w_{main}*S_{off-main}+w_{opt}*S_{off-opt}$$
(29)$$S_{on-final}=w_{main}*S_{on-main}+w_{opt}*S_{on-opt}$$
where $S_{off-final}$ and $S_{on-final}$ denote the offline violence detection score and the online violence detection score.

### 3.4. Loss Function

#### 3.4.1. Online Prediction Loss

To reduce the difference between online prediction and ground truth *y*, we use the BCE loss to calculate the training loss as follows:(30)$$L_{on}=-\frac{1}{n}\overset{N}{\underset{i=1}{\sum }}\left(y^{i}\ln\left({\hat{y}}_{on}^{i}\right)+\left(1-y^{i}\right)\ln\left(1-{\hat{y}}_{on}^{i}\right)\right)$$
where *N* denotes the batch size, and $y^{i}$ and ${\hat{y}}_{on}^{i}$ denote the ground truth *y* and the output of online prediction, respectively.

#### 3.4.2. Offline Prediction Loss

Similarly, the loss function of the offline prediction module is as follows:(31)$$L_{off}=-\frac{1}{n}\overset{N}{\underset{i=1}{\sum }}\left(y^{i}\ln\left({\hat{y}}_{off}^{i}\right)+\left(1-y^{i}\right)\ln\left(1-{\hat{y}}_{off}^{i}\right)\right)$$
where ${\hat{y}}_{off}^{i}$ denotes the output of offline prediction.

#### 3.4.3. Cross Prediction Loss

To reduce the difference between the online prediction and the offline prediction, we use L2 loss to calculate the training loss as follows:(32)$$L_{on-off}=\overset{N}{\underset{i=1}{\sum }}{∥{\hat{y}}_{off}^{i}-{\hat{y}}_{on}^{i}∥}^{2}$$

#### 3.4.4. Total Loss

Finally, the total loss function is the weighted sum of the above items as follows:(33)$$\begin{array}{c} L=L_{off-main}+L_{on-main}+\varepsilon L_{on-off-main}+L_{off-opt}+L_{on-opt}+\delta L_{on-off-opt} \end{array}$$
where ε and δ denote the hyperparameter that controls the importance of the L2 loss, and herein, we set ε = 5 and δ = 3. $F_{off-main}$ denotes the loss function of the offline prediction module in the main branch, and $F_{off-opt}$ denotes the loss function of the offline prediction module in the optical flow branch.

## 4. Experiments

### 4.1. Datasets

XD-Violence [22] is the only existing violence detection dataset including audio, optical flow, and RGB modals, and it is the largest-scale public multi-modal dataset, having a total of 217 h at present. Among them, the training dataset contains 3954 videos, and the test dataset contains 800 videos. It is collected from CCTV cameras, hand-held cameras, car driving recorders, etc. It provides more than eight scenarios and six types of violent events, and each violent video includes multiple violent labels $\left(1\leq \mathrm{labels}\leq 3\right)$. As shown in Figure 3, the feature extraction module [22] includes two branches: visual and audio. The visual branch extracts RGB features and optical flow features from the I3D [43] network, and the audio branch extracts audio features from the VGGish [44,45] network.

### 4.2. Evaluation Criteria

Similar to the previous methods [22,26], we use frame-level average precision (AP) as the evaluation criteria on the XD-Violence dataset. Average precision can evaluate the quality of the proposed model in classification, and a larger AP value indicates better performance.

### 4.3. Implementation Details

Our OAMFN method is implemented based on PyTorch 1.8.1, an NVIDIA Tesla T4 GPU with 16 GB of memory. During the training stage, our method uses the Adam optimizer [46], a learning rate of 0.0001, a batch size of 64, and 30 epochs. The dropout function in the prediction module has a dropout rate of 0.3. Parameters and in the offline detection module are set to 1. Parameters μ and θ in the offline detection module are set to 1.

### 4.4. Ablation Study: Comparison of Modules in Our Method

To explore the effects of each individual module on the AP performance in the OAMFN, we employ the XD-Violence dataset as an example, and perform an ablation study to evaluate their influence on violence detection. As shown in Table 1, when utilizing the CIFM, the CAM, and the OASFM separately, the AP result achieves only 80.69%, 81.39%, and 80.8% on the XD-Violence dataset, respectively. The combination of the CIFM and the OASFM improves the AP performance to 81.87% and exceeds the utilization of one of them alone. When using the combination of the CAM and the OASFM, the AP result increases to 82.43%. When both the CIFM and the CAM are added, the AP performance is boosted to 82.58%. These results indicate that the combination of these modules can further improve the AP performance over using one of them alone, indicating that our modules motivate each other. When combining all three modules, the AP result increases to 83.09%, which is the best performance. This shows that these three modules are complementary, and that the combination of all three modules is valid for increasing the difference between violence and non-violence videos. In particular, in order to prove the effectiveness of the OASFM, we compare it with another fusion strategy, namely “non-score fusion”. For non-score fusion, we concatenate different kinds of modalities, such as RGB–optical flow, optical flow–audio, and audio–RGB, to form three branches. These three branches are concatenated to be one main branch before entering the channel attention module. Using the optical flow-aware weighting mechanism, the AP performance is 0.51% higher than that with “non-score fusion”. This shows that the proposed optical flow-aware score fusion mechanism can effectively control the contributions of the main branch and the optical flow branch under different optical flow conditions and boost the AP performance of violence detection.

### 4.5. A Comparison of the AP Performance with the Existing Methods on the XD-Violence Dataset

We compare the XD-Violence dataset with the current unsupervised and semi-supervised methods. As shown in Table 2, our OAMFN method is superior to the current unsupervised method in offline detection. In offline detection, compared with weakly supervised methods, our method is 9.89%, 4.45%, 5.28%, and 1.4% higher than Sultani et al. [8], Wu et al. [22], Tian et al. [47], and Pang et al. [26], respectively. For the online detection task, our method is 4.42% higher than Wu et al. [22]. Hence, our experimental results indicate that the OAMFN can increase the difference between violence and non-violence videos, and it is efficient for integrating optical flow, RGB, and audio modals with the task of violence detection.

### 4.6. A Comparison of the Offline AP Performance on the Different Violent Classes

To verify the AP performance of our method on all kinds of violence videos, we select 30 videos from each of the 6 violent classes (i.e., abuse, car accident, explosion, fighting, riot, and shooting) on the XD-Violence dataset for testing. As shown in Figure 4, our OAMFN method is superior to that of Wu et al. [22] in five violent classes, evidently lifting the AP performance by 6.43% to 22.03% in violent classes with large changes in optical flow, such as car accidents and explosions. Compared with the AP performance of our main branch (RGB and audio features as input) and Wu et al. [22], the AP performance of our non-score fusion framework shows that the addition of optical flow features has effectively improved the detection accuracy, and our optical flow-aware-based score fusion strategy can further improve the detection accuracy in five violent classes (i.e., abuse, car accident, explosion, fighting, and riot). Our OAMFN method is slightly lower in the shooting class because, in videos with weak action or short duration, the change in optical flow is very small. Comparing our main branch with our non-score fusion framework, the AP result shows that the addition of optical flow features reduces the detection accuracy. However, our optical flow-aware-based score fusion strategy improves the detection accuracy by 4% after controlling the contributions of three modals. Hence, our experimental results indicate that the addition of optical flow features can improve violent classes with poor AP performance, such as car accidents and explosions. Additionally, the optical flow-aware-based score fusion strategy can further improve the detection accuracy.

### 4.7. Qualitative Results

Figure 5 demonstrates the violence score curves produced by our OAMFN method in different testing videos from the XD-Violence dataset. We select six violent classes (i.e., explosion, abuse, car accident, fighting, shooting, and riot) on the XD-Violence dataset for testing. As shown in Figure 5, the rise in violence scores means the emergence of violence. Our method can clearly separate violent fragments from non-violent fragments in five violent classes (i.e., car accident, explosion, fighting, riot, and shooting). Our OAMFN method is less effective in terms of the AP performance of the abuse class because this class shows no obvious violence in the audio modal, the optical flow modal, and the RGB modal.

## 5. Conclusions

In this paper, we found that optical flow features play a certain role in identifying specific violent behaviors, such as car accidents and explosions, which shows the lowest performance of AP among the six violence categories; thus, we added optical flow features and designed a score weighting mechanism based on optical flow awareness to control the impact of optical flow features which makes this research innovative and evidently lifts the AP performance by 6.43% to 22.03% in violent classes with large changes in optical flow, such as car accidents and explosions. Moreover, adding optical flow information can extract the motion features of objects and effectively solve the problems of short duration and weak action in the task of violence detection. We propose a novel two-branch optical flow-aware-based multi-modal fusion network for violence detection, which integrates audio features, the optical flow features, and the RGB features into a unified framework. First, the main branch concatenates RGB features and audio features and the optical flow branch concatenates optical flow features with RGB features and audio features, respectively. Then, the cross-modal information fusion module integrates the features of different combinations and applies weights to them to capture cross-modal information in audio and video. After that, the channel attention module extracts valuable information by weighting the integration features. Furthermore, an optical flow-aware-based score fusion strategy is introduced to fuse features of different modalities from two branches. However, there are still two challenging problems to be solved. First, as shown in Figure 5, our method has difficulty with accurately determining the boundary of violence. Second, our model is not an end-to-end model. An obvious disadvantage is that the training objectives of each module are inconsistent, and thus, the trained system has difficulty achieving optimal performance in the end. In contrast, the end-to-end model can avoid the above problems and reduce the complexity of the project.

## Figures and Tables

**Figure 1 entropy-24-00939-f001:**
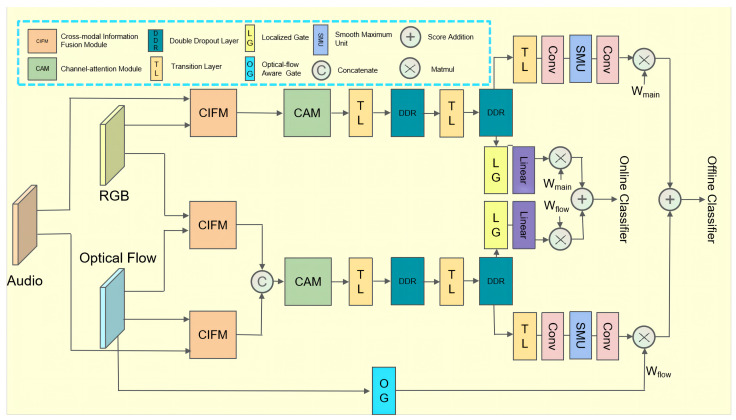
Our proposed OAMFN. The cross-modal information fusion module for capturing the cross-modal information and fusing multi-modal features, as well as the channel attention for meaningful information selection and the prediction module for predicting score generation and fusing two branches via the optical flow-aware-based score fusion strategy.

**Figure 2 entropy-24-00939-f002:**
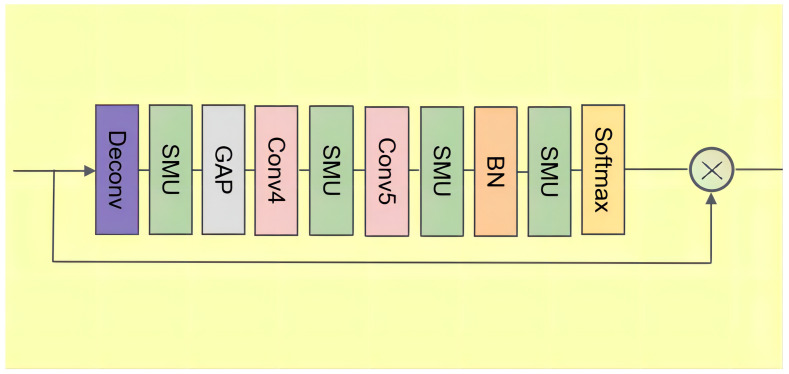
Structures of the channel attention.

**Figure 3 entropy-24-00939-f003:**
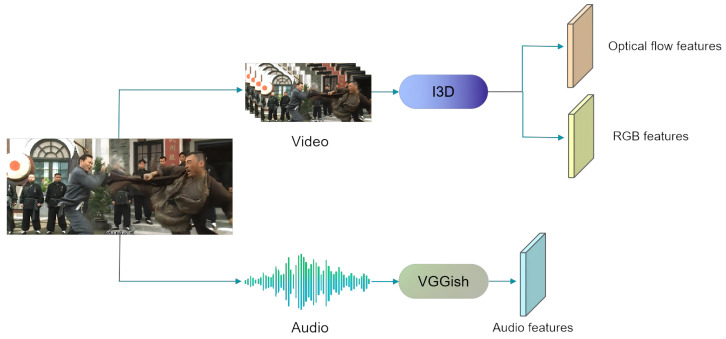
Schematic diagram of the feature extraction module.

**Figure 4 entropy-24-00939-f004:**
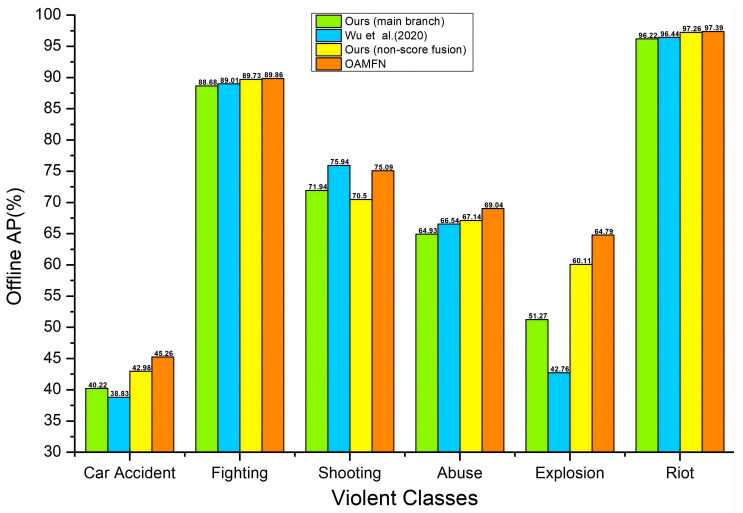
A comparison of the offline AP performance with Wu et al. [22] on the violent classes.

**Figure 5 entropy-24-00939-f005:**
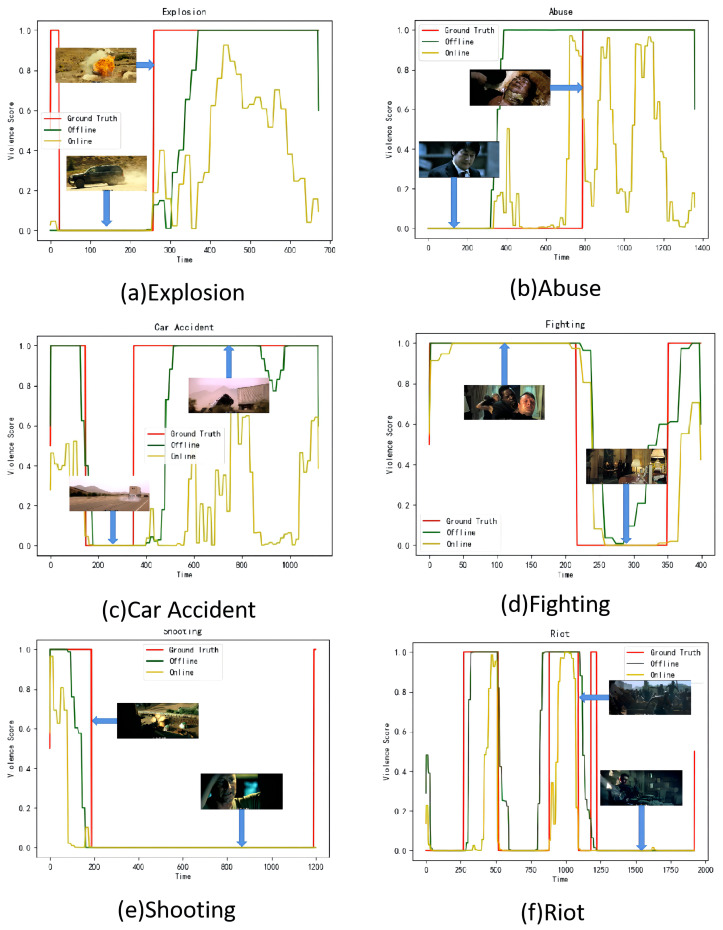
Qualitative results in the testing videos from the XD-Violence dataset.

**Table 1 entropy-24-00939-t001:** Ablation study: a comparison of the modules in our method.

Cross-Attention	Channel Attention	Optical Flow-Aware Fusion	AP (%)
🗸			80.69
	🗸		81.39
		🗸	80.8
🗸	🗸		82.58
🗸		🗸	81.87
	🗸	🗸	82.43
🗸	🗸	🗸	83.09

**Table 2 entropy-24-00939-t002:** A comparison of the AP performance with the existing methods on the XD-Violence dataset. The best results are in *red* and the second-best results are in *blue*.

Supervision	Method	Feature	Online AP(%)	Offline AP(%)
Unsupervised	SVM	-	-	50.78
	OCSVM [48]	-	-	27.25
	Hasan et al. [49]	-	-	30.77
Weakly Supervised	Sultani et al. [8]	RGB	-	73.2
	Wu et al. [22]	RGB + Audio	73.67	78.64
	Tian et al. [47]	RGB	-	77.81
	CRFD [24]	RGB	-	75.90
	Pang et al. [26]	RGB + Audio	-	81.69
	MSL [25]	RGB	-	78.59
	Ours (without OASFM)	RGB + Flow + Audio	77.24	82.58
	Ours (with OASFM)	RGB + Flow + Audio	78.09	83.09

## Data Availability

Our training set XD-Violence datasets can be obtained from: https://roc-ng.github.io/XD-Violence/ (accessed on 6 July 2020).

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
