# Peer review of "Optical Flow-Aware-Based Multi-Modal Fusion Network for Violence Detection"

_entropy, 2022, doi:10.3390/e24070939_

Round 1

Reviewer 1 Report

The authors present a novel method for violence detection utilizing both visual and audio information and a novel network architecture. The results show an improvement over other techniques recently published in the literature. While the text is relatively clear, the figures are difficult to interpret. Figures 5-10 specifically are illegible to the reader and need to be improved. Meanwhile Figure 4 is comparatively quite large and probably should be made smaller. 

My main technical concern is that it is unclear how statistically significant the improvement is over other techniques. The authors cite ~1.5% and ~4.5% improvements in the performance which seems small given the added complexity of the model. How would such a fractional increase in performance result in realizable gains by an end user? I think this question needs to be addressed before publication can be considered.

Author Response

Point 1: The authors present a novel method for violence detection utilizing both visual and audio information and a novel network architecture. The results show an improvement over other techniques recently published in the literature. While the text is relatively clear, the figures are difficult to interpret. Figures 5-10 specifically are illegible to the reader and need to be improved. Meanwhile Figure 4 is comparatively quite large and probably should be made smaller.

Response 1: We are very grateful to Reviewer for reviewing the paper so carefully. We are very sorry for our negligence of the explanation. See figure 4 and figures 5(a)-5(f) in attachment PDF for specific changes.

Point 2: My main technical concern is that it is unclear how statistically significant the improvement is over other techniques. The authors cite ~1.5% and ~4.5% improvements in the performance which seems small given the added complexity of the model. How would such a fractional increase in performance result in realizable gains by an end user? I think this question needs to be addressed before publication can be considered.

Response 2: Thank you for your comments. The introduction part of the original manuscript of our paper does not clearly describe the new idea and importance; In view of this, we have strengthened the introduction to emphasize the innovation. The innovation of this paper lies in we found that optical flow features play a certain role in identifying specific violent behaviors, such as car accident and explosion which shows the lowest performance of AP among the six violence categories, so we added optical flow features and designed a score weighting mechanism based on optical-flow aware to control the impact of optical flow features which makes this research innovative and evidently lifting the AP performance by 6.43% to 22.03% in violent classes with large changes in optical flow, such as car accidents and explosions. We believe that our ideas on the promotion of these two types of violence can be used as a reference for other scholars on violence detection. In addition, our model can achieve real-time detection, improve the detection accuracy of violence at the expense of a little speed, and reduce the harm. Such gains and losses are acceptable to us.

Reviewer 2 Report

In this paper, the authors introduce a network structure of optical-flow aware based multi-modal fusion module for violence detection. We found that optical flow features play a certain role in identifying specific violent behaviors, so we added optical flow features and designed a score weighting mechanism based on optical-flow aware to control the impact of optical flow features which makes this research innovative. Also, we divided the data set into six categories according to different categories of violence, thus the limitations of the experiment can be analyzed. Compared with other related methods, our results show better performance. However, there are some problems in this paper, which need to be greatly improved. Some of my comments are as follows:

1.In Section 4.5, 9 comparative methods are listed, and some of them have not been described and analyzed in the Introduction section.

2.As shown in Table 2, the proposed model and competing methods achieve experiment on XD-Violence dataset. To verify the proposed method, a comparison should be performed on other datasets.

3.Labels of Fig.5 to Fig.10 are written in a wrong way. Please modify them and mark the corresponding representation.

4. In order to make the conclusion section more clear, authors are highly encouraged to include the point-by-point findings of this article. How the proposed work can be used to support the emerging applications of major industry, including intelligent medical care, intelligent transportation, smart ocean, and intelligent manufacturing?

5.The authors need to double check all typos and grammatical errors. Another point is that in the References section, some of the authors use full names, some of the authors use their initials. Please check and keep consistent.

Author Response

Point 1: In Section 4.5, 9 comparative methods are listed, and some of them have not been described and analyzed in the Introduction section. 

Response 1: Thank you for your comments. The introduction part of the original manuscript of our paper does not write some comparison algorithms clearly; In view of this, we have strengthened the introduction to emphasize the comparison algorithm.

Point 2: As shown in Table 2, the proposed model and competing methods achieve experiment on XD-Violence dataset. To verify the proposed method, a comparison should be performed on other datasets.

Response 2: We agree that more study or more data would be useful to verify our proposed method. But XD-Violence is the only existing large-scale dataset offering visual and audio data of violent events at the same time, we will explain the lack of datasets in Section 4.2. Also, we divided the dataset into six categories according to different categories of violence, thus the limitations of the experiment can be analyzed.

Point 3: Labels of Fig.5 to Fig.10 are written in a wrong way. Please modify them and mark the corresponding representation.

Response 3: We are very grateful to Reviewer for reviewing the paper so carefully. We are very sorry for our negligence of the explanation. See figures 5(a)-5(f) in attachment PDF for specific changes.

Point 4:  In order to make the conclusion section more clear, authors are highly encouraged to include the point-by-point findings of this article. How the proposed work can be used to support the emerging applications of major industry, including intelligent medical care, intelligent transportation, smart ocean, and intelligent manufacturing?

Response 4:  Thank you for your comments. The conclusion part can make the reader understand the work of this article clearly; In view of this, we have strengthened the conclusion section to emphasize the point-by-point findings of this article. Also, we have strengthened the introduction section to emphasize applications of violence detection.

Point 5: The authors need to double check all typos and grammatical errors. Another point is that in the References section, some of the authors use full names, some of the authors use their initials. Please check and keep consistent.

Response 5: We are very sorry for our incorrect writing and it is rectified at the References section.

Round 2

Reviewer 1 Report

Thank you for addressing my comments. The axis labels on Figure 5 are still illegible. Please fix these.